# The role of halogens in Au−S bond cleavage for energy-differentiated catalysis at the single-bond limit

Peihui Li[1,5], Songjun Hou[2,5], Qingqing Wu[2], Yijian Chen[1], Boyu Wang[1], Haiyang Ren[1], Jinying Wang[1], Zhaoyi Zhai [3], Zhongbo Yu [3] ✉, Colin J. Lambert [2] ✉, Chuancheng Jia[1] ✉ & Xuefeng Guo [1,4] ✉

The transformation from one compound to another involves the breaking and formation of chemical bonds at the single-bond level, especially during catalytic reactions that are of great significance in broad fields such as energy conversion, environmental science, life science and chemical synthesis. The study of the reaction process at the single-bond limit is the key to understanding the catalytic reaction mechanism and further rationally designing catalysts. Here, we develop a method to monitor the catalytic process from the perspective of the single-bond energy using high-resolution scanning tunneling microscopy single-molecule junctions. Experimental and theoretical studies consistently reveal that the attack of a halogen atom on an Au atom can reduce the breaking energy of Au−S bonds, thereby accelerating the bond cleavage reaction and shortening the plateau length during the single-molecule junction breaking. Furthermore, the distinction in catalytic activity between different halogen atoms can be compared as well. This study establishes the intrinsic relationship among the reaction activation energy, the chemical bond breaking energy and the single-molecule junction breaking process, strengthening our mastery of catalytic reactions towards precise chemistry.

Catalysis can reduce the activation energy of a reaction and speed up chemical and biological reaction processes. It plays a key role in energy conversion[1], environmental science[2], life science[3], chemical synthesis[4], and other fields[5–7]. The study of the catalytic reaction process at the single-bond limit, especially the exploration of the single-bond breaking process that determines the occurrence of the reaction, is the key to the development of catalyst design and synthetic chemistry.

Especially, halogens play an important role in promoting metal catalysis[8–10]. However, the exact mechanism remains unclear.

Over the past decades, single-molecule technologies[11,12] have been developed to monitor the reaction process at the single-molecule level using optical, mechanical, and electrical signals[5,13–16]. For instance, the reaction process of a single enzyme can be observed through single-molecule fluorescence tracking[17], and the vibration information of a

[1]Center of Single-Molecule Sciences, Institute of Modern Optics, Frontiers Science Center for New Organic Matter, Tianjin Key Laboratory of Micro-Scale Optical Information Science and Technology, College of Electronic Information and Optical Engineering, Nankai University, 38 Tongyan Road, Jinnan District, 300350 Tianjin, People's Republic of China. [2]Department of Physics, Lancaster University, Lancaster LA1 4YB, UK. [3]State Key Laboratory of Medicinal Chemical Biology, College of Pharmacy, Nankai University, 300350 Tianjin, People's Republic of China. [4]Beijing National Laboratory for Molecular Sciences, National Biomedical Imaging Center, College of Chemistry and Molecular Engineering, Peking University, 292 Chengfu Road, Haidian District, 100871 Beijing, People's Republic of China. [5]These authors contributed equally: Peihui Li, Songjun Hou. ✉e-mail: zyu@nankai.edu.cn; c.lambert@lancaster.ac.uk; jiacc@nankai.edu.cn; guoxf@pku.edu.cn

single chemical bond can be obtained through single-molecule Raman detection[18,19]. Atomic force microscopy (AFM)[20], and optical/magnetic tweezers can detect mechanical information of single molecules[21]. Furthermore, the species produced during the reaction can be imaged at the single-bond level by scanning tunneling microscopy (STM) and non-contact atomic force microscopy (NC−AFM)[22]. Based on single-molecule conductance monitoring, single-molecule junction technologies can be used for chemical analysis[23–27], especially for detecting the intermediates and products formed, as well as the kinetics of the reactions[28–31]. To obtain more details about catalytic reactions at the single-molecule level, especially the single-bond breaking energy that determines the activation energy of the reaction, a method to study chemical reactions at the single-bond limit in an in-situ environment needs to be developed.

In a single-molecule break junction, the stretch-induced breaking process of single-molecule junctions is related to the breaking energy of a chemical bond between the electrode atom and the molecular terminal atom. Here, we used single-molecule break junction measurements to study the role of halogens in metal catalysis and explore the effect of catalysis on single-bond breaking by monitoring the stretch-induced breaking process, especially the corresponding plateau length in relation to the breaking energy. The gold-sulfur (Au−S) bond is chosen as a simple model system, which has broad implications in molecular biology[32], surface science[33,34], and molecular electronics[35]. The Au−S bond breaking energy, which corresponds to the activation energy of the Au−S bond cleavage reaction, is expected to reduce under halogen catalysis (Fig. 1). For instance, according to theoretical calculations, both chloride (Cl) (Supplementary Fig. 1) and iodine (I) (Supplementary Fig. 2) catalysis can reduce the activation energy for Au−S bond cleavage. Specifically, in the process of Au−S bond breaking, the Au−S bond breaking energy will decrease as the halogen atom attacks the Au atom. After that, the halogen atom will depart from Au to realize catalyst regeneration. As the energy required is equivalent to the difference between the Au−S breaking energy without and with halogen catalysis, only a small activation energy is required for each step of the reaction. On this basis, the relationship among the reaction activation energy, the chemical bond breaking energy and the measured plateau length can be established to describe the effect of halogen catalysis on the breaking of single Au−S bonds, so as to realize the exploration of the essence of the catalytic reaction.

## Results

### Monitor the Au−S bond-breaking process

Specifically, a π-conjugated molecule (2,6-bis(((4-acetylthio)phenyl) ethynyl) antracene) with acetylthiol (−SAc) terminal groups, denoted as AC−SAc (Supplementary Fig. 3), is used for the study. The S atom at the end of the AC−SAc molecule and the Au atom on the gold electrode

will form an Au−S bond, which will break during the stretching process. To investigate this stretching process with Au−S bond breaking, scanning tunneling microscopy break junction (STM−BJ) experiments were performed. Specifically, when the tip is lifted from the gold substrate, the single-molecule junction could form with a molecule connecting to the gold tip and substrate through Au−S bonds (Fig. 2a). As the tip moves away, the molecule will be stretched until it reaches a critical point (state "1"). Then, if the tip continues to lift, the Au−S bond will break (state "2"), resulting in a rapid drop in conductance. During this stretching process, a key factor is the plateau length, which can be observed from a single trace and plateau-length statistics, defined as the distance the junction can be stretched before the Au−S bond breaks. In general, the plateau length is related to the molecular length and interfacial interactions. When the molecule-electrode interaction is weak, the interfacial bonding energy mainly determines the plateau length. It should be noted that the plateau length in this study has not been corrected for the Au−Au snap-back distance ($\approx$0.5 nm)[36]. If the molecular length needs to be determined by the plateau length, the Au−Au snap-back distance needs to be added back.

To measure the intrinsic properties of Au−S bond cleavage of the AC−SAc molecule, experiments were first carried out in a halogen-free dodecane solvent. The conductance of the single-molecule junction has been repeatedly measured as a function of tip-substrate displacement to establish a conductance-displacement relationship. Typical traces for the junctions under 0.1 V bias are shown in Fig. 2b (blue) and Supplementary Figure 4. Specifically, the conductance features of integer multiples of $G_O$ ($G_O = 2\,e^2/h$) can be observed firstly. Then, after the Au atomic contact is broken, the conductance shows rapid tunneling decay. Afterward, a conductance plateaus corresponding to the conductance of the target molecule can be found at a specific value (Supplementary Fig. 4c, green). Corresponding to the stretching process shown in Fig. 2a, the symbols "1" and "2" in Fig. 2b have the same meaning to Fig. 2a, representing the "going to break" and "just broken" states of the Au−S bond during the stretching process, respectively. Thousands of the conductance traces are used to construct the two-dimensional (2D) conductance-displacement histograms (Fig. 2b) of the AC−SAc molecule (Supplementary Fig. 4a). A Lorentzian fitting conductance peak at $\approx$10$^{-4.5}$ $G_O$ ($\approx$2.451 nS) can be observed from 2D conductance-displacement histograms (Supplementary Fig. 4b). In addition, the plateau length is defined from 10$^{-0.3}$ $G_O$ down to 10$^{-1}$ below the specific conductance peak, corresponding to a Lorentzian fitting peak at $\approx$0.97 nm in statistics (Fig. 2c), reflecting the interface interaction of Au−S bonds. The experiment of the AC−SAc in mesitylene (TMB) solution has also been done, and the results are consistent with the case in dodecane solution (Supplementary Fig. 5), illustrating that the different plateau lengths are independent on solvent viscosity.

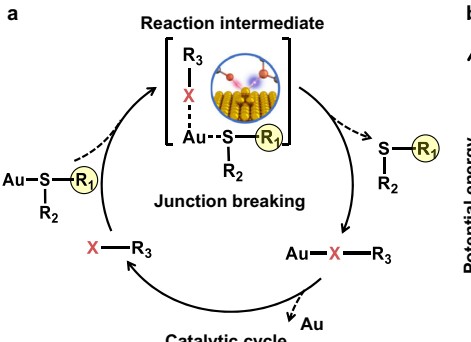
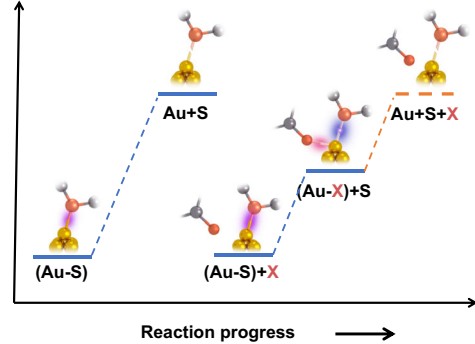

**Fig. 1 | Overview of halogen atom-assisted catalysis. a** Schematic diagrams for Au−S bond breaking with halogen catalysis. **b** Schematic energy diagrams for Au−S bond breaking without and with halogen catalysis. X is the halogen atom. The orange color represents the halogen atom coming from solvent; the yellow color represents the Au atom; the pink color represents the sulfur atom; the gray color represents the carbon atom.

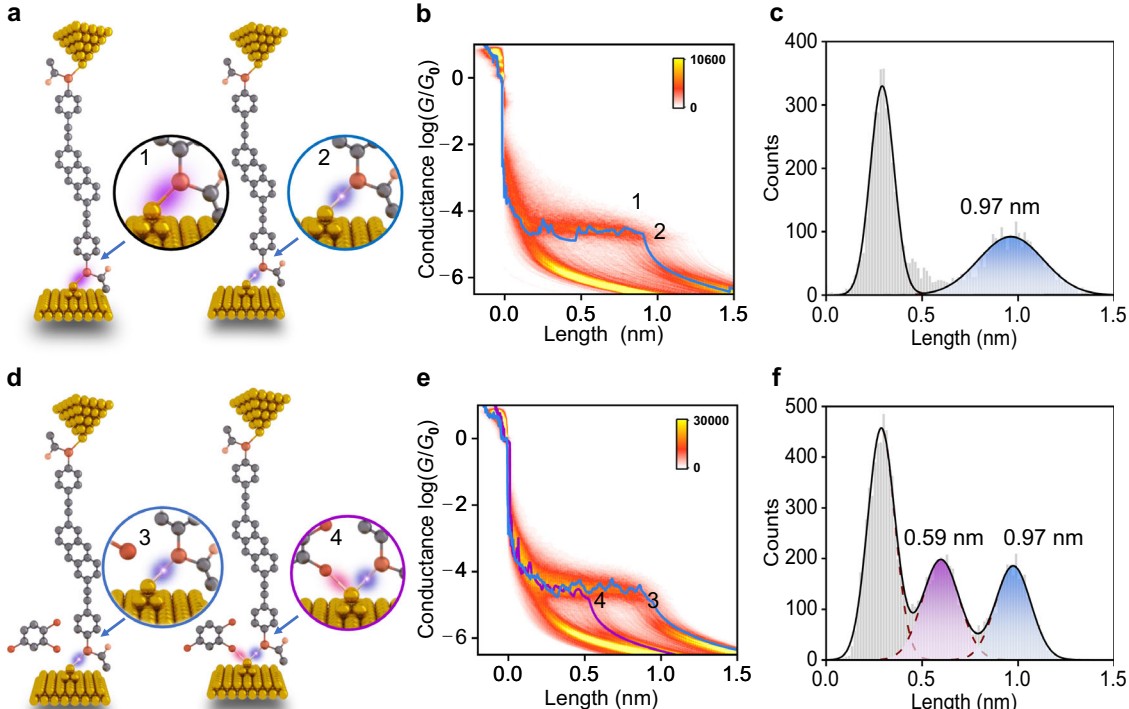

**Fig. 2 | Measurements of AC−SAc single-molecule break junctions. a** Schematic illustration of the Au−S bond cleavage reaction in STM−BJs. The left panel shows the formation of a single-molecule junction; the right panel shows force-induced breaking of the junction at the interface. **b** 2D conductance-displacement histograms for junctions in dodecane, consisting of ≈4600 individual traces. The blue line is a typical single trace corresponding to the length peak in **c**. The labels "1" and "2" represent the "going to break" and "just broken" states of the Au−S bond during the stretching process. **c** Statistics of conductance plateau lengths for junctions in dodecane. **d** Schematic illustration of two cases of Au−S bond breaking in TCB. **e** 2D conductance-displacement histograms for junctions in TCB, consisting of ≈10,000 individual traces. The purple line with a short plateau and the blue line with a long plateau are two typical single traces corresponding to the two length peaks in **f**. The labels "3" and "4" represent the junction breaking without and with chlorine attack. **f** Statistics of conductance plateau lengths for junctions in TCB. The blue color represents the case in the absence of Cl. The purple color represents the case in the presence of Cl.

To study the catalytic reaction process of Au−S bond cleavage in the presence of halogen (Fig. 2d), 1,2,4-trichlorobenzene (TCB) with chlorine (Cl) atoms was selected as the catalyst, which is also a commonly used solvent in STM−BJ experiments (Supplementary Figs. 6 and 7). Therefore, experiments with the AC−SAc molecule in the TCB solution environment were carried out. In 2D conductance-displacement histograms and single traces (Fig. 2e), two characteristic signals with different plateau lengths can be observed. Consistently, two plateau lengths of ≈0.59 nm and ≈0.97 nm corresponding to the Lorentzian fitting peaks can also be obtained from the plateau-length statistics (Fig. 2e, f), which represent the two cases of Au−S bond cleavage. The longer plateau length is consistent with that of the AC−SAc molecule in dodecane, indicating that the TCB acts only as a solution and has no interaction with the bonded Au atom on the electrode ("3"). In contrast, the shorter plateau length means that the Au−S bond interaction is weak, indicating that the breaking process of the Au−S bond is accelerated due to the interaction between Au and Cl ("4"). It should be noted that when the acetyl-protecting groups of the AC−SAc molecules are deprotected by tetrabutylammonium hydroxide (TBA−OH) to thiols (named as AC−SH), there is only one characteristic plateau length observed (Supplementary Fig. 8). Its conductance is about half an order of magnitude higher than that of AC−SAc (Supplementary Fig. 9).

To determine the generality of this phenomenon, another π-conjugated molecule (1,4-bis(((4-acetylthio)phenyl)ethynyl)benzene) with acetylthiol terminal groups, noted as OPE3-SAc, was also investigated. During the single-molecule junction breaking experiments in TCB, two plateau-length peaks can also be observed, located at ≈0.48 nm and ≈0.69 nm, respectively (Supplementary Figs. 10 and 11). This excludes the effect of the molecular conjugate skeleton, further demonstrating that chlorine atoms can be used as catalysts to accelerate the breaking of Au−S bonds.

## Accelerate the breaking of Au−S bonds by chlorine

Density functional theory (DFT) calculations were carried out to reveal the underlying mechanism behind the observations in experiments. The junction evolution of AC−SAc is investigated by successively pulling the top electrode to 20 Å with an interval of 0.2 Å (Fig. 3a, b), and fully relaxing the target molecules and the two gold pyramid tips at each step while freezing the other part of gold electrodes. Then, the energy of the whole junction at each step is extracted, and the force is evaluated via the derivative of energy with respect to the displacement of the tip. As guided by the blue arrow in Fig. 3b, the force goes down significantly signaling the rupture of Au−S bonds as illustrated in the final image of Fig. 3a (denoted as 4 in Fig. 3a, b). The discontinuities shown in the energy-displacement curve originate from significant geometry changes between two adjacent steps (Supplementary Fig. 12) after relaxation. This rupture force is around 0.8 nN smaller than the 1.4 nN force required to break the single-atomic gold contact as reported in the literature[37].

The effect of the halogen atom (e.g., Cl) is investigated by placing TCB close to the gold tip and then pulling the measured molecule away by 0.2 Å at each step until 2 Å. In this process, the TCB is fully relaxed, while freezing the measured molecule to mimic the real pulling process in experiments. During the gradual stretching process of Au−S bonds, the energy increases until the measured molecule is pulled

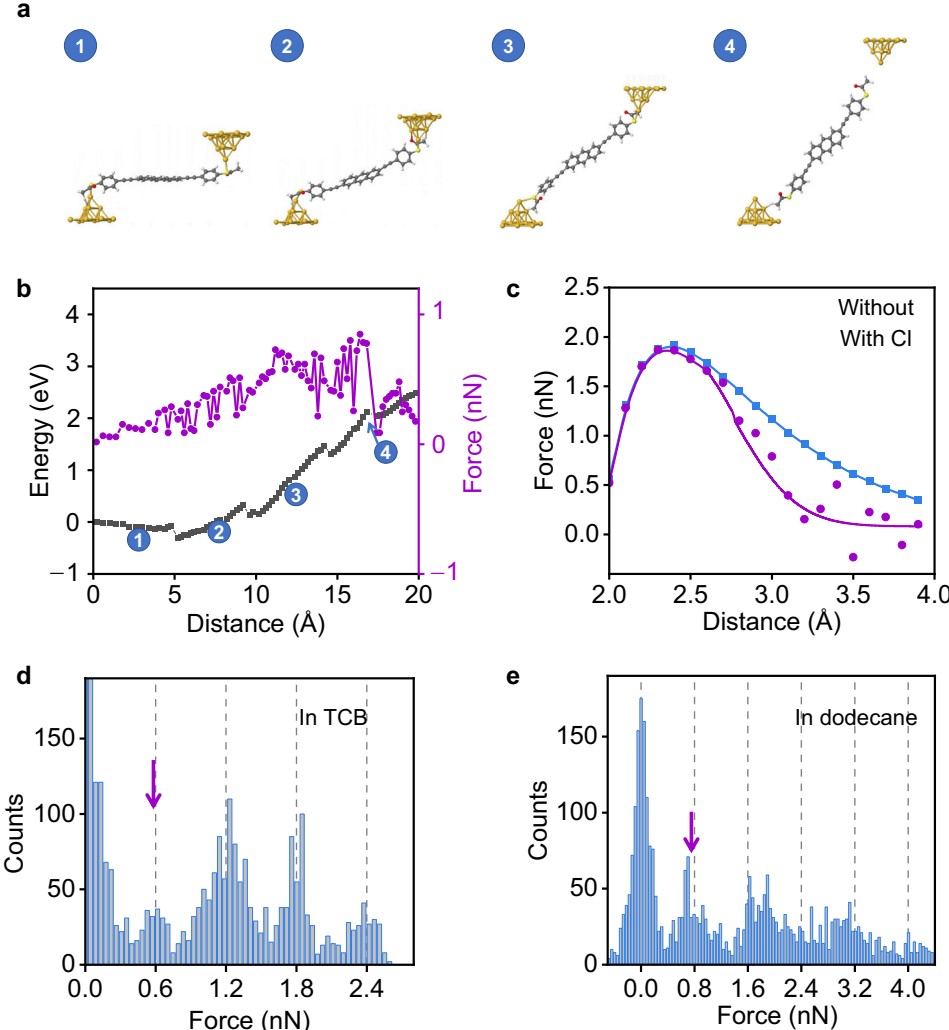

**Fig. 3 | Stretching process of a single Au−S bond. a** Theoretical simulated stretching processes of an AC−SAc single-molecule junction. **b** Corresponding theoretical predictions for the total energy ("square") and force evolution ("circle") as stretching for −SAc anchors using the initial structure as a reference. **c** Theoretical predictions for the force evolution of the AC−SAc molecule as stretching with/without TCB, marked with "circle" and "square", respectively. **d** Force histograms for OPE3−SAc in TCB; **e** Force histograms for OPE3−SAc in dodecane. The blue color represents the case in the absence of Cl. The purple color represents the case in the presence of Cl.

away. It can be observed when the chlorine in TCB acts on the gold atom, the force needed is slightly lower than that without chlorine (Fig. 3c). This is also supported by experimental single-molecule force spectroscopy measurements by using AFM (Fig. 3d, e and Supplementary Fig. 13). As results showed in Fig. 3d, e, the force required to break the −SAc anchored single molecule from the Au substrate in TCB is ≈0.6 nN, which is lower than ≈0.8 nN in dodecane in absence of Cl. In addition, in comparison with the absence of the halogen atom, when Cl is close to Au, the energy needed to break Au−S bonds decreases from ≈1.5 eV to ≈1.0 eV (Fig. 4a and Supplementary Fig. 14). A more direct characterization of the strength changes of Au−S bonds caused by the halogen atom is to monitor the change of the lowest value of the electron density ($\rho_{min}$) along the axis between Au and S[38]. It can be observed that $\rho_{min}$ decreases from ≈0.3102 to ≈0.3097 a.u. as moving Cl close to the Au atom from ≈6 Å to ≈3 Å, indicating that the presence of Cl in TCB weakens Au−S bonds (Fig. 4b).

**The universality of halogen catalysis in breaking Au−S bonds**
Another halogen atom iodine is also considered in our work, which is included in triiodobenzene (TIB) solvent. When iodine is employed, the energy required to break the Au−S bond is reduced further to

≈0.5 eV (Fig. 4a). $\rho_{min}$ is decreased further to ≈3.092 due to the stronger interaction between Au and I in comparison with Cl (Fig. 4b), demonstrating that iodine has a better performance than chlorine in accelerating the rupture of Au−S bonds. Correspondingly, the stronger catalytic acceleration effect of iodine on Au−S bond cleavage was also investigated experimentally. Here, TIB is selected as the control catalyst. Specifically, TIB was dissolved in TCB at a concentration of 3 mM. In the STM−BJ experiment, AC−SAc single-molecule junctions in the TIB/TCB solution show three distinct plateau lengths of ≈0.46 nm, ≈0.64 nm, and ≈0.92 nm, corresponding to the Lorentzian fitting peaks (Fig. 4c, d). The long and middle plateau lengths are consistent with the results for the AC−SAc junction in the TCB solution. That is, the long plateau length of ≈0.92 nm originates from the breaking of Au−S without any interference, while the intermediate plateau length of ≈0.64 nm is due to the interaction between Au and Cl. At the same time, a new short plateau length of ≈0.46 nm appears, which is due to the breaking of the Au−S bond under the action of I. This experiment proves that iodine can be used as a more efficient catalyst to reduce the breaking energy of Au−S bonds and accelerate the Au−S bond cleavage reaction. It should be noted that the different surface diffusivities of I and Cl on the gold surface are likely to be a factor affecting

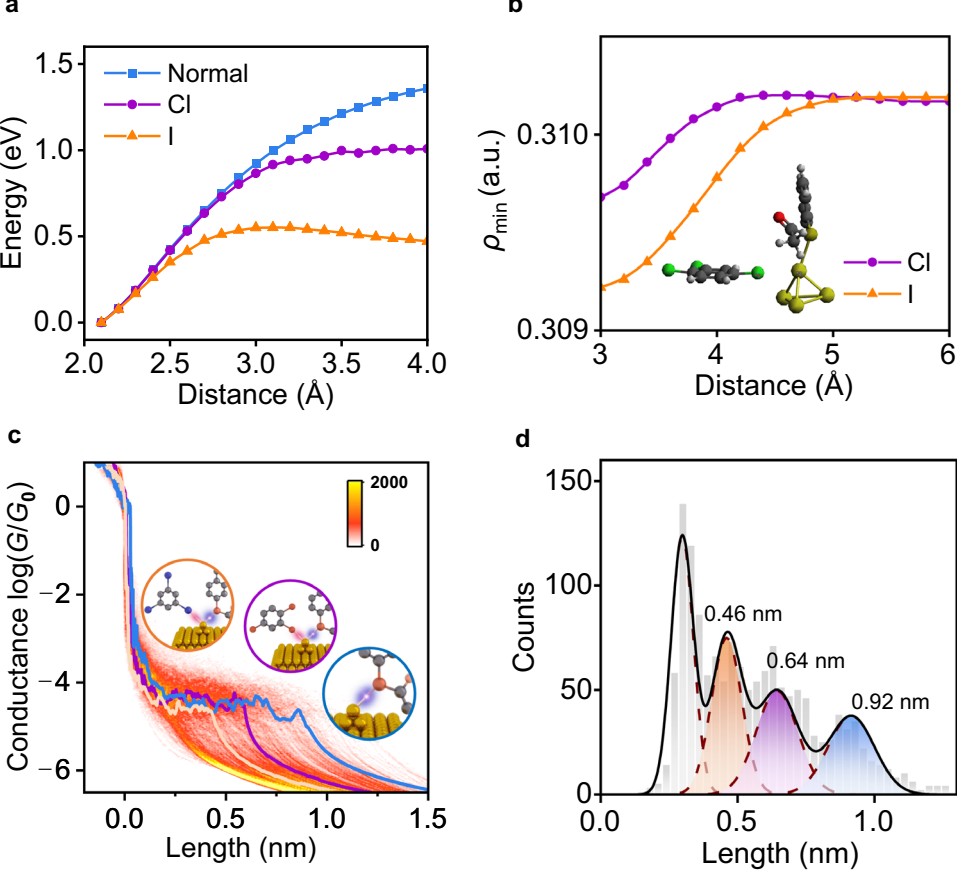

**Fig. 4 | Comparison of the Cl⁻ and I-catalyzed single Au–S bond cleavage.**
**a** Energy evolution during Au–S bond cleavage in the absence and presence of Cl or I catalysis with the energy of optimal position as a reference. **b** Changes of the lowest value of $\rho_{min}$ along the axis between Au and S as a function of the distance between Au and Cl or I. The unit is atomic units (a.u.). **c** 2D conductance-displacement histograms for the junctions in TIB/TCB, consisting of ≈2500 individual traces. The orange/purple/blue lines are three typical single traces with

different plateau lengths. Insets: schematic illustration of the corresponding Au–S bonds in different cases. The orange/purple/blue circles represent the cases with I, with Cl, and without I or Cl, respectively. **d** Statistics of conductance plateau lengths for junctions in TIB/TCB. The blue color represents the case in the absence of Cl and I. The purple color represents the case in the presence of Cl. The orange color represents the case in the presence of I.

the lifetime of the junctions[39,40] (Supplementary Fig. 15). This difference is reflected in the different plateau lengths in Fig. 4c. Since the junction is immersed in the solution, the solvents should surround the gold tips formed during the pulling process (see Fig. 3a). The surrounding halogen compound solvents can bind to the gold tip directly and impact the junction lifetime. Afterward, they can diffuse along the electrode surface from the gold tip. It is expected that the larger iodine can bind for a longer time with the tip to disrupt the splitting of Au–S due to its higher diffusion barrier on the gold surface, leading to short plateau lengths and highly efficient catalysis performance. Similarly, bromine can also be used as a catalyst to reduce the breaking energy of Au–S bonds. This results in three distinct plateau lengths of ≈0.50 nm, ≈0.64 nm, and ≈0.98 nm for the single-molecule junction experiment of AC–SAc in 1,2,4-tribromobenzene (TBB)/TCB solution (Supplementary Fig. 16).

In summary, we have developed an approach to study the catalytic reaction process from the perspective of the single-bond energy using a single-molecule break junction technique. The relationship among the reaction activation energy, the chemical bond breaking energy and the measured plateau length during junction breaking is established to study the reactions. Combining experimental and theoretical studies, it is found that the attack of the halogen atom on the Au atom can reduce the potential energy of Au–S bond cleavage during the bond-breaking process, thereby accelerating the reaction. In addition, the difference in the catalytic activity of iodine and

chlorine catalysts for Au–S bond cleavage can also be monitored. This provides a promising way to explore the catalytic reaction mechanism at the single-bond level, which is helpful for catalyst design and synthetic chemistry, and the rich electrochemistry of halogens on metals, which is one of the simplest models for electrochemically driven adsorption. It is worthwhile to mention that since organo-metallic clusters/complexes generally have relatively large band gaps (Supplementary Fig. 17), there should be plenty of room to couple single-molecule measurements with optical measurements.

## Methods
### Experimental preparation
The gold wire (99.99%, 0.25 mm diameter) was purchased from ZhongNuo Advanced Material (Beijing) Technology Co., Ltd. for the fabrication of the STM tip. The gold tip was obtained by electrochemical corrosion. Substrates were prepared by depositing a 10 nm thick chromium film and a 200 nm thick gold film on the N monocrystalline face of a silicon wafer. The TCB (Aladdin) purity is ≈99% as stated by the supplier; the dodecane (Aladdin) is ≈98% as stated by the supplier; the TMB (Aladdin) is ≈97% as stated by the supplier. The sample preparation: the target molecules were mixed with the corresponding solvent into a solution.

In the STM−BJ measurement, the distance between the gold tip and the substrate was controlled by a stepper motor and a piezo stack. The bias voltage was applied between the tip and substrate, and the

current was used as the feedback to control the movement of the gold tip. During the repeating opening (tip retracting) and closing (tip approaching) cycles, the conductance versus displacement traces were collected, and the traces of the opening cycles were used for further analysis. All measurements were carried out at room temperature. The single-molecule conductance measurements were carried out using Xtech STMBJ from Xiamen University, and the data was analyzed by XMe open-source code (https://github.com/Pilab-XMU/XMe_DataAnalysis).

## Single-molecule force spectroscopy measurement

Single-molecule force spectroscopy measurements were carried out by repeatedly moving the AFM tip into and out of contact with a gold substrate in a solution containing sample molecules (Multimode 8, Bruker Co.). The process was controlled by a feedback loop (home-made electronics) that started by driving an AFM tip into contact with the substrate at a rate of 206 nm s$^{-1}$. The AFM tip (NPG-10) was purchased from Ruideyi Technology (Wuhan) Co., LTD. The spring constant of the AFM cantilever was calibrated to be 0.3638 N m$^{-1}$.

## Theoretical methods

Geometrical optimizations were performed using the DFT code SIESTA[41], with a local density approximation LDA functional, a double-$\zeta$ polarized basis, a cutoff energy of 200 Ry, and a 0.02 eV Å$^{-1}$ force tolerance. In order to compute their conductance, the molecules were each placed between pyramidal Au electrodes. The optimal distance between the Au tip and the S atom was relaxed to be around 2.4 Å (Supplementary Figs. 18 and 19). For each structure, the transmission coefficient $T(E)$ describing the propagation of electrons of energy $E$ from the left to the right electrodes was calculated using Gollum code[42], which combines the mean-field Hamiltonian and overlap matrices of the DFT code SIESTA with the Landauer-based quantum transport theory. This is equivalent to using the expression:

$$T(E) = tr[\Gamma_L(E)G_r(E)\Gamma_R(E)G_r + (E)] \tag{1}$$

Where tr means the trace of the matrix, $\Sigma_{L,R}(E) = i(\Sigma_{L,R}(E) - \Sigma_{L,R}^+(E))/2$, $G_r(E) = (g^{-1} - \Sigma_L - \Sigma_R)^{-1}$, and g is the Green's function of the isolated molecule. $\Gamma_{L,R}$ determines the widths of transmission resonances. $\Sigma_{L,R}(E)$ are the self-energies describing the contact between the molecule and left (L) or right (R) electrodes. While $G_r$ is the retarded Green's function of the molecule in the presence of the electrodes.

Figures 3c and 4a were computed within the Vienna Ab initio simulation package (VASP 5.4.4)[43,44]. The projector augmented wave (PAW) pseudopotentials[45] and the Perdew–Burke–Ernzerhof (PBE) scheme[46] for the exchange and correlation energy are employed. The energy cutoff is set to 400 eV. A $20 \times 20 \times 25$ Å$^3$ supercell and the $\Gamma$-point were used.

It should be noted that the force in Fig. 3c is at least twice as large as that in Fig. 3b, because different junctions are employed. Specifically, as shown in Fig. 3a, the molecule and the two gold pyramid tips are allowed to relax at each pulling step, which leads to Fig. 3b. In Fig. 3c, only one electrode is used and frozen. Furthermore, the target molecule is also simplified to one phenyl ring.

## Reporting summary

Further information on research design is available in the Nature Portfolio Reporting Summary linked to this article.

## Data availability

The data that support the findings of this study are available from the corresponding authors upon request. Raw data is available via *Zenodo*[47].

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

## Acknowledgements
We acknowledge Prof. Ryan C. Chiechi and Dr. Marco Carlotti for providing molecular samples. We acknowledge primary financial support from the National Key R&D Program of China (2021YFA1200101 and 2022YFE0128700 to X.G., and 2021YFA1200102 to C.J.), the National Natural Science Foundation of China (22150013 and 21933001 to X.G., and 22173050 to C.J.), the Natural Science Foundation of Beijing (2222009) to X.G., the New Cornerstone Science Foundation through the XPLORER PRIZE to X.G., "Frontiers Science Center for New Organic Matter" at Nankai University (63181206) to X.G., and the Natural Science Foundation of Beijing (2222009) to X.G. the Fundamental Research Funds for the Central Universities (63223056) to C.J., the Beijing National Laboratory for Molecular Sciences (BNLMS202105) to C.J., the support from the Leverhulme Trust for Early Career Fellowship ECF-2023-757 to S.H., the financial support from the UK EPSRC, through grant nos. EP/M014452/1, EP/X026876/1, EP/P027156/1 and EP/N03337X/1 to C.J.L.

## Author contributions
X.G., C.J., and C.J.L. conceived the idea for the paper. P.L. and Y.C. carried out the experimental measurements. S.H. and Q.W. built and analyzed the theoretical model and performed the quantum transport calculation. Z.Y. and Z.Z. provided single-molecule force spectroscopy equipment and technical support. X.G., C.J., C.J.L., Z.Y., P.L., S.H., Q.W., B.W., H.R., Y.C., and J.W. analyzed the data and wrote the paper. All the authors discussed the results and commented on the manuscript.

## Competing interests
The authors declare no competing interests.
