## [Peer Review File · Nature Communications]

The role of halogens in Au-S bond cleavage for energy-differentiated catalysis at the single-bond limitREVIEWER COMMENTS

Reviewer #1 (Remarks to the Author):

Most definitely a well-researched paper and an instructive reading. In my humble opinion however the title and abstract (and probably to some degree also the introduction and conclusions) do not make justice to the main contribution of the paper: a new and very insightful measurements of interactions between halogens and metals. The role of halogens in chemical catalysis, a notable example is the remarkable increase in selectivity towards epoxidation catalysed by silver when a small amount of chlorine is present, has been known for about 150 years, used in industry, but poorly understood. Adding STM single molecule measurements to the list of tools available to study halogen interactions with metals (and probably also semiconductors) is a big step forward. Halogens on metals also have a very rich electrochemistry, and are one of the simplest models for electrochemically driven adsorption. So once again I can see a lot of follow up studies from this initial report.

I would also like a bit more "speculations" in the conclusion section when it comes to what can follow this first report. For example halogen-metal complexes have generally semi-conductive properties, I believe with relatively large band gaps, and there could be space to couple these single molecule measurements with optical measurements?

In terms of mechanism, do the authors believe that there is a barrier to the initial adsorption? And after this event, is there likely to be an electron transfer between halogen and metal? Is there a possibility of diffusion of the halogen compounds along the surface? Will this impact the lifetime of the junction? From memory I believe the lateral diffusion of halogens on Cu(111) is faster for the larger halogens. Would this help explain the data in Figure 4?

Could the counts be added to the y-axis of Figure 4d (and a similar comment for Figures 2 and 3)?

The forces mentioned in the manuscript are actual forces or output of calculations? If they are experimental numbers, could the authors expand on how they were measured?

Figure 1. Transient state. Is this a depiction of the transition state or of a reaction intermediate?

Reviewer #2 (Remarks to the Author):

In this research, Li and colleagues delve into the breaking of individual bonds and the reduction of bond-breaking energy. Employing a combined experimental and theoretical approach, they investigate single-molecule junctions in a scanning tunneling microscope (STM) along with density functional theory (DFT). Specifically, the study focuses on the breaking of a Sulfur-gold (S-Au) bond while considering the presence of solvent molecules. The experimental setup involves suspending a single molecule in the tip-substrate junction of the STM, and the conductance is measured as a function of separation. Interestingly, the presence of halogenated solvent molecules is found to influence the S-Au bond energy. Notably, in a non-halogenated solvent environment, the single-molecule junction exhibits greater stability compared to a chlorinated solvent setting. Furthermore, introducing a solvent containing iodine leads to further reductions in the junction's stability. Supporting the experimental findings, DFT calculations are performed with the corresponding solvent positioned in close proximity to the S-Au bond in the junction, revealing a similar trend in S-Au stability. The authors hypothesize that an interaction between halogen and Au atoms weakens the S-Au bond. The researchers also demonstrate the robustness of their observations by employing two different molecules with distinct backbones in the junction, indicating that the observed trend is a general phenomenon rather than being reliant on a specific molecule type. These findings shed light on the intriguing interplay between solvent molecules and the breaking energy of single bonds in the context of single-molecule junctions. The interpretation of the data is consistent between experiment and simulation and in accord with the authors' hypothesis.

A few points might be addressed before publishing the manuscript.

The discussion of Figures 3 and 4 is not linear in the text, i.e. the reader is forced to repeatedly jump between the figures to make sense of the text. Restructuring the figures or cleaning up the discussion would be highly appreciated.

Page 9, line 181: '... to break Au-S bonds decrease from ~ 1.2 eV to ~ 1 eV (Fig. 4a).' Where in the Figure is the point of bond breaking? The blue line keeps rising above 1.2 eV. Is the energy taken at a fixed distance?

Page 9, lines 182-183: The authors go on to conclude 'when the chlorine in TCB is acted on the gold atom, the force needed is lower than that without chlorine (Fig. 3c)' Yet both curves show a maximum force required of ~ 2.0 nN that needs to be overcome. Maybe a more detailed description of how these curves were calculated will help. Also, a comment on why the force in Fig. 3c is at least twice as large compared to Fig. 3b might be helpful.

More information on the orientation of the solvent molecule with respect to the Au atom in DFT calculations would help to understand the halogen-Au interaction. In Supplementary Fig. 10 it does not appear that there is a direct halogen-Au interaction, but the perspective might be misleading. Typical distances between Cl, I and the topmost Au atom also would be interesting to judge the strength of the interaction.

In Fig. 3 of the main text, where do the discontinuities in the energy-distance dependence originate from? Is this a computational artifact?

Reviewer #3 (Remarks to the Author):

I read this paper with a lot of interest. The authors studied the Au-S bond breaking reaction under halogen catalysis using single molecule junctions. A relationship among the reaction activation energy, the chemical bond breaking energy and the measured junction plateau length was established elegantly. The paper is written very well and easy to follow. The use of single molecule junctions for these type of catalysis reactions is novel and hence I feel the paper is suitable for Nature Comm. I, however, have few technical inquiries:

- I am wondering why was not mesitylene used instead of the halogen free dodecane solvent? It appears to me that mesitylene would be a better control molecule than dodecane and more comparable to TCB and TIB.
- Since the authors are using Au-Au junctions and the plateau length is a central to the study, I am wondering if the authors have corrected for Au-Au snap-back distance (~ 0.5 nm) which should be added to the plateau length?
- Also related to the previous question, the authors used different solvents of different viscosities. Did the author consider that different viscosities could lead to different Au-Au snap back distances, which in turn can affect the plateau lengths? A discussion about this is needed.
- Is there a reason why 1,2,4-tribromobenzene was not used like TIB as the catalyst to strengthen the argument?
- The references list is perhaps a bit brief. I recommend extending the list to include recent papers and reviews that uses single molecule junctions for chemical analysis.

- Is there a possibility that the thioesters are reduced to thiols during the measurements? Would that affect the measurements as potentially one could have a mixture of thiols and thioesters. A brief discussion would help.

All in all, this is an interesting and a clever study.

Listed below are the major changes in the new version of the manuscript.

1. We have reorganized the content in the introduction and conclusion sections, adding the role of halogens in chemical catalysis and the corresponding literature as Refs. [8–10].

Please see Pages 2, 3, and 12 in the revised main text.
2. We have added the discussion about the semiconductive properties of halogen-metal complexes to conclusion.

Please see Page 12 in the revised main text and Page S15 in the revised Supporting Information.
3. We have added the calculated energy and Voronoi charge distribution when gradually changing the distance between Cl and gold atom and the corresponding description to discuss the diffusion of the halogen compounds along the surface.

Please see Page 10 in the revised main text.
4. We have cited the literature Refs. [39, 40] to support that there should be diffusion of the halogen compounds along the gold surface.

Please see Page 10 in the revised main text.
5. We have revised Figures 2–4 and added the specific counts to the y-axis of Figure 4d, Figure 2c, f and Figure 3d, e.

Please see Pages 7, 8, and 11 in the revised main text.
6. We have taken the experimental method of force out from “Experimental preparation”, as “Single-molecule force spectroscopy measurement” part of “Methods”.

Please see Page 15 in the revised main text.
7. We have used "Reaction Intermediate" to replace "Transient State" in Figure 1a.

Please see Page 4 in the revised main text.
8. We have rearranged the discussion of Figures 3 and 4 in the revised main text.

Please see Page 9 in the revised main text.
9. We have added the energy evolution during the Au–S bond cleavage in the absence of halogen catalysis with a larger range.

Please see Page 9 in the revised main text and Page S13 in the revised Supporting Information.

10. We have added the details and description about the calculation of force.
Please see Pages 9, 16, and 17 in the revised main text.
11. We have added the calculated model details as Fig. S19.
Please see Page S16 in the revised Supporting Information.
12. We have added the explanation about the origin of the discontinuities in the energy-displacement curve.
Please see Page 8 in the revised main text and Page S11 in the revised Supporting Information.
13. We have done the STM–BJ experiment with AC–SAc in TMB solution and added the results as Figure S5.
Please see Page 5 in the revised main text and Page S5 in the revised Supporting Information.
14. We have added the discussion of the Au–Au snap-back distance and corresponding reference as Ref. [36].
Please see Page 5 in the revised main text.
15. We have added the control experiment of AC–SAc in TBB/TCB solvent as Figure S16.
Please see Page 10 in the revised main text and Page S14 in the revised Supporting Information.
16. We have added the literature about the single molecule junctions using for chemical analysis as Refs. [23–27].
Please see Page 3 in the revised main text.
17. We have added the experimental results of AC–SH in TCB solvent as Figure S8 and the calculated transmission spectra of –SAc and –SH anchors as Figure S9.
Please see Page 6 in the revised main text and Page S8 in the revised Supporting Information.

Reviewer 1:

General comments: Most definitely a well-researched paper and an instructive reading. In my humble opinion however the title and abstract (and probably to some degree also the introduction and conclusions) do not make justice to the main contribution of the paper: a new and very insightful measurements of interactions between halogens and metals. The role of halogens in chemical catalysis, a notable example is the remarkable increase in selectivity towards epoxidation catalysed by silver when a small amount of chlorine is present, has been known now for about 150 years, used in industry, but poorly understood. Adding STM single molecule measurements to the list of tools available to study halogen interactions with metals (and probably also semiconductors) is a big step forward. Halogens on metals also have a very rich electrochemistry, and are one of the simplest models for electrochemically driven adsorption. So once again I can see a lot of follow up studies from this initial report.

General Reply: We thank the reviewer very much for his/her highly affirmative and pertinent comments, constructive suggestions, and kind support. We have combined the reviewer's original comments together and revised the manuscript fully according to his/her suggestions. Especially, inspired by the reviewer, we reorganized the content in the introduction and conclusion sections, and added the role of halogens in chemical catalysis and corresponding references. In order to respond clearly, we divided them and made specific replies for each opinion as shown below. After revisions, the manuscript has been significantly strengthened.

Our Revision: We reorganized the content in the introduction and conclusion sections by adding the role of halogens in chemical catalysis and the following sentence in the revised main text: "Especially, halogens play an important role in promoting metal catalysis⁸⁻¹⁰. However, the exact mechanism remains unclear." (Page 2); "Here, we used single-molecule break junction measurements to study the role of halogens in metal catalysis" (Page 3); "This provides a promising way to explore the catalytic reaction mechanism at the single-bond level, which is helpful for catalyst design and synthetic chemistry, and the rich electrochemistry of halogens on metals, which is one of the simplest models for electrochemically driven adsorption." (Page 12). The corresponding literature has been added to the main text as Refs. [8-10] (Page 2).

The following references have been added accordingly:

Ref. [8] Wang, T. et al., Halogen-incorporated Sn catalysts for selective electrochemical

CO₂ reduction to formate. *Angew. Chem. Int. Ed.* **62**, e202211174 (2023).

Ref. [9] Górski, B. et al., Copper-catalysed amination of alkyl iodides enabled by halogen-atom transfer. *Nat. Catal.* **4**, 623–630 (2021).

Ref. [10] Mahmudov, K.T. et al., Noncovalent interactions in metal complex catalysis. *Coord. Chem. Rev.* **387**, 32–46 (2019).

Comment 1: I would also like a bit more “speculations” in the conclusion section when it comes to what can follow this first report. For example, halogen-metal complexes have generally semi conductive properties, I believe with relatively large band gaps, and there could be space to couple these single molecule measurements with optical measurements?

Our Reply: Thanks a lot for the good comment. We agree with the reviewer. Organo-metal clusters/complexes generally have relatively large band gaps, which are evidenced by optimal measurements (*J. Am. Chem. Soc.*, **2015**, *138*, 390). We also performed spin-polarized calculations to investigate the HOMO-LUMO gaps of two halogen-metal complexes as shown below. HOMO-LUMO gaps of ~0.79 eV and ~1.35 eV were obtained.

Our Revision: We have added the following sentence to the conclusion section in the revised main text: “It is worthwhile to mention that since organo-metallic clusters/complexes generally have relatively large band gaps, there should be plenty of room to couple single-molecule measurements with optical measurements.” (Page 12). We have added the HOMO-LUMO gap results as Figure R1 (Fig. S17) in the revised Supplementary Information. The corresponding description has been added to the Supplementary Information as follows: “The organo-metallic clusters/complexes generally have relatively large band gaps, which can be proved by optimal measurements^{S2} and our simulation (Fig. S17)” (Page S15).

The following references have been added accordingly:

Ref. [S2] Abbas, M. A. *et al.*, Exploring interfacial events in gold-nanocluster-sensitized solar cells: Insights into the effects of the cluster size and electrolyte on solar cell performance. *J. Am. Chem. Soc.* **138**, 390–401 (2015).

Fig. R1 (Fig. S17). The HOMO-LUMO gaps of two examples (relaxed halogen-gold complexes). Five gold atoms with one chlorine and four gold atoms with two chlorines are analyzed, respectively.

Comment 2: In terms of mechanism, do the authors believe that there is a barrier to the initial adsorption? And after this event, is there likely to be an electron transfer between halogen and metal? Is there a possibility of diffusion of the halogen compounds along the surface? Will this impact the lifetime of the junction? From memory, I believe the lateral diffusion of halogens on Cu(111). Would this help explaining the data in Figure 4?

Our Reply: Thanks a lot for the good comment. To clarify this point, the initial adsorption process is studied by gradually changing the distance between Cl and gold atoms. As shown in Figure R2 (Fig. S15), the energy decreases continually until an optimal distance ($\sim 2.3 \text{ \AA}$), suggesting that there is no barrier for the whole adsorption process. Meanwhile, the Voronoi charge distribution is employed to evaluate the charge transfer between Cl and Au atoms. The positive charge means a loss of electrons. As Cl atom approaches Au atom, the Cl starts to transfer electron to the Au atom. We found that around $0.1 e^-$ is transferred from Cl atom to Au atom at the optimal distance.

As expected by the reviewer, there should be diffusion of the halogen compounds along the gold surface. A literature (*J. Am. Chem. Soc.* **2013**, *135*, 5768) about the diffusion process of bromine and iodine atoms on Au (111), Ag (111) and Cu (111) demonstrated that the larger iodine indeed diffuses faster than bromine on Cu (111) and Ag (111) (Energy barrier: 60 meV vs 70 meV), while it is slower on Au (111) (110 meV vs 90 meV) (see Figure R3 from the reference: *J. Am. Chem. Soc.* **2013**, *135*, 5768). The higher barrier for iodine on the gold surface is also confirmed in a review paper (*Surf. Sci. Rep.* **2018**, *73*, 83), where the energy barriers are found to be $\sim 20 \text{ meV}$, ~ 90

meV and ~110 meV for chlorine, bromine and iodine, respectively. Therefore, the kinetics of the diffusion process depend on the specific metal surface. The higher energy barrier for iodine diffusion on the gold surface means that iodine stays with a gold atom for a longer time, thus providing further support to our demonstration that iodine has a better catalytic performance in accelerating the Au–S single bond splitting.

Our Revision: We have added the calculated energy and Voronoi charge distribution when gradually changing the distance between Cl and gold atoms as Figure R2 (Fig. S15). We have also added the following sentences to discuss the diffusion of the halogen compounds along the surface: “It should be noted that the different surface diffusivities of I and Cl on the gold surface are likely to be a factor affecting the lifetime of the junctions^{39,40} (Supplementary Fig. 15). This difference is reflected in the different plateau lengths in Figure 4c. Since the junction is immersed in the solution, the solvents should surround the gold tips formed during the pulling process (see Figure 3a). The surrounding halogen compound solvents can bind to the gold tip directly and impact the junction lifetime. Afterwards, they can diffuse along the electrode surface from the gold tip. It is expected that the larger iodine can bind for a longer time with the tip to disrupt the splitting of Au–S due to its higher diffusion barrier on the gold surface, leading to short plateau lengths and highly efficient catalysis performance.” (Page 10). The corresponding literature has been added to the main text as Refs. [39 and 40].

The following references have been added accordingly:

Ref. [39] Andryushechkin, B.V. *et al.*, Adsorption of halogens on metal surfaces. *Surf. Sci. Rep.* **73**, 83–115 (2018).

Ref. [40] Björk, J. *et al.*, Mechanisms of halogen-based covalent self-assembly on metal surfaces. *J. Am. Chem. Soc.* **135**, 5768–5775 (2013).

Fig. R2 (Fig. S15). Evolution of energy and Voronoi charge as the distance between gold tip atoms and chlorine atoms. The optimal distance between gold and halogen atoms is indicated by the red dash line. The positive charge means a loss of electrons.

Fig. R3. Energy diagram for the diffusion of bromine (a) and iodine (b) on the coinage metal surfaces: Cu (111), Ag (111) and Au (111). The energy is given as the total energy with respect to the FCC hollow site (IS). The bridge site is a transition state (TS) on all surfaces. (c) Illustration of the reaction pathway for bromine on Au (111). Reprinted (adapted) with permission from Björk, J. et al., Mechanisms of halogen-based covalent self-assembly on metal surfaces. *J. Am. Chem. Soc.* 135, 5768–5775 (2013). Copyright {2013} American Chemical Society.

Comment 3: Could the counts be added to the y-axis of Figure 4d (and a similar comment for Figures 2 and 3)?

Our Reply: Thanks a lot for the good comment. We have added the specific counts to the y-axis of Figure 4d (Fig. R4), Figure 2c, f (Fig. R5) and Figure 3d, e (Fig. R6).

Our Revision: We have revised Figures 2-4 and added the specific counts to the y-axis of Figure 4d, Figure 2c, f and Figure 3d, e in the revised main text.

Fig. R4 (Fig. 4d). The counts have been added to the y-axis in Figure 4d.

Fig. R5 (Fig. 2). The counts have been added to the y-axis in Figure 2c and 2f.

Fig. R6 (Fig. 3). The counts have been added to the y-axis in Figure 3d and 3e.

Comment 4: The forces mentioned in the manuscript are actual forces or output of calculations? If they are experimental numbers, could the authors expand on how they were measured?

Our Reply: Thanks a lot for the good comment. The forces mentioned in the manuscript include both the actual forces and the output of calculations. Specifically, the force results of Figures 3b, c and Figure 4a originate from the calculations, while the force results of Figures 3d, e originate from AFM experiments. The forces in Figures 3b, c are calculated via the formula dE/dx , where dE is the change of energy and dx is the displacement of the tip. Both are extracted from first principles DFT calculations. As for the experimental method of force, we have introduced the details in the “Experimental preparation” of the “Methods” section in the main text (Page 15) and Supplementary Fig. 13 in the Supporting Information (Page S12). Specifically, single-molecule force spectroscopy measurements were carried out by repeatedly moving the AFM tip into and out of contact with a gold substrate in a solution containing sample molecules (Multimode 8, Bruker Co.). The process was controlled by a feedback loop (home-made electronics) that started by driving an AFM tip into contact with the substrate at a rate of 206 nm/s. The AFM tip (NPG-10) was purchased from Ruideyi Technology (Wuhan) Co., LTD. The spring constant of the AFM cantilever was calibrated to be ~ 0.3638 N/m. To make it easier for readers to find, we take it out from “Experimental preparation” as the “Single-molecule force spectroscopy measurement” part of “Methods” (Page 15).

Our Revision: To make it easier for readers to find, we have taken the experimental method of force out from “Experimental preparation” as the “Single-molecule force spectroscopy measurement” part of “Methods” (Page 15).

Comment 5: Figure 1. Transient state. Is this a depiction of the transition state or of a reaction intermediate?

Our Reply: Thanks a lot for the good comment. It is a reaction intermediate. Since a pulling force is acting on the tip during the reaction process, it is challenging to identify a transition state, which in principle is calculated without external force.

Our Revision: To clarify this point, we have used "Reaction Intermediate" to replace "Transient State" in Figure 1a (Fig. R7) (Page 4).

Fig. R7 (Fig. 1a). "Transient State" has been replaced by "Reaction Intermediate".

Reviewer 2:

General comments: In this research, Li and colleagues delve into the breaking of individual bonds and the reduction of bond-breaking energy. Employing a combined experimental and theoretical approach, they investigate single-molecule junctions in a scanning tunneling microscope (STM) along with density functional theory (DFT). Specifically, the study focuses on the breaking of a Sulfur-gold (S–Au) bond while considering the presence of solvent molecules. The experimental setup involves suspending a single molecule in the tip-substrate junction of the STM, and the conductance is measured as a function of separation. Interestingly, the presence of halogenated solvent molecules is found to influence the S–Au bond energy. Notably, in a non-halogenated solvent environment, the single-molecule junction exhibits greater stability compared to a chlorinated solvent setting. Furthermore, introducing a solvent containing iodine leads to further reductions in the junction's stability. Supporting the experimental findings, DFT calculations are performed with the corresponding solvent positioned in close proximity to the S–Au bond in the junction, revealing a similar trend in S–Au stability. The authors hypothesize that an interaction between halogen and Au atoms weakens the S–Au bond. The researchers also demonstrate the robustness of their observations by employing two different molecules with distinct backbones in the junction, indicating that the observed trend is a general phenomenon rather than being reliant on a specific molecule type. These findings shed light on the intriguing interplay between solvent molecules and the breaking energy of single bonds in the context of single-molecule junctions.

The interpretation of the data is consistent between experiment and simulation and in accord with the authors' hypothesis. A few points might be addressed before publishing the manuscript.

General Reply: We thank this reviewer very much for his/her professional comments, constructive suggestions and kind support. According to these comments, we have added more evidence and details, and made a comprehensive revision of the manuscript. After revisions, the manuscript has been significantly strengthened.

Comment 1: The discussion of Figures 3 and 4 is not linear in the text, i.e. the reader is forced to repeatedly jump between the figures to make sense of the text. Restructuring the figures or cleaning up the discussion would be highly appreciated.

Our Reply: Thanks a lot for the good comment. We apologize for the disorganized structure of the discussion, which impacts the flow of reading. According to the suggestion of the reviewer, we have rearranged the discussion of Figures 3 and 4 in the revised main text (Page 9).

Our Revision: We have rearranged the discussion of Figures 3 and 4 in the revised main text, as shown below (Page 9): “It can be observed when the chlorine in TCB act on the gold atom, the force needed is slightly lower than that without chlorine (Fig. 3c). This is also supported by experimental single-molecule force spectroscopy measurements by using AFM (Fig. 3d,e and Supplementary Fig. 13). As results showed in Figs. 3d and 3e, the force required to break the –SAC anchored single molecule from the Au substrate in TCB is ~ 0.6 nN, which is lower than ~ 0.8 nN in dodecane in absence of Cl. In addition, in comparison with the absence of the halogen atom, when Cl is close to Au, the energy needed to break Au–S bonds decrease from ~ 1.5 eV to ~ 1.0 eV (Fig. 4a and Fig. S14). A more direct characterization of the strength changes of Au–S bonds caused by the halogen atom is to monitor the change of the lowest value of the electron density (ρ_{min}) along the axis between Au and S³⁸. It can be observed that ρ_{min} decreases from ~ 0.3102 to ~ 0.3097 a.u. as moving Cl close to the Au atom from ~ 6 Å to ~ 3 Å, indicating that the presence of Cl in TCB weakens Au–S bonds (Fig. 4b).”

Comment 2: Page 9, line 181: ‘... to break Au–S bonds decrease from ~ 1.2 eV to ~ 1 eV (Fig. 4a).’ Where in the Figure is the point of bond breaking? The blue line keeps rising above 1.2 eV. Is the energy taken at a fixed distance?

Our Reply: Thanks a lot for the reviewer’s comments. Yes, at each step, the distance between the tip and the measured molecule is fixed, while the TCB is allowed to relax to mimic the pulling process in the experiments. The exact point of bond breaking is hard to identify, since the energy is increasing continuously. We thank the reviewer for noting that the blue curve of Fig. 4a keeps rising at a distance of 4 Å. To clarify the trend after this point, we continue pulling the tip until the distance reaches ~ 10 Å. As shown in Figure R8 (Fig. S14), the energy keeps increasing to ~ 1.55 eV at a distance of 6 Å, suggesting that the presence of a halogen atom causes further energy to decrease because the red curve is already flat.

Our Revision: We have modified this sentence in Page 9 of the main manuscript to “... to break Au–S bonds decrease from ~1.5 eV to ~1.0 eV (Fig. 4a and Fig. S14)” and added the following figure as Fig. S14.

Fig. R8 (Fig. S14). Energy evolution during the Au–S bond cleavage in the absence of halogen catalysis. The energy needed to break the Au–S bond is guided by the red dash line.

Comment 3: Page 9, lines 182-183: The authors go on to conclude ‘when the chlorine in TCB is acted on the gold atom, the force needed is lower than that without chlorine (Fig. 3c)’ Yet both curves show a maximum force required of ~2.0 nN that needs to be overcome. Maybe a more detailed description of how these curves were calculated will help. Also, a comment on why the force in Fig. 3c is at least twice as large compared to Fig. 3b might be helpful.

Our Reply: Thanks a lot for the reviewer’s comments. Indeed, the distance between the tip and the measured molecule is fixed, while the TCB is allowed to relax to mimic the pulling process in the experiments. The forces in Fig. 3c are calculated via the formula $F = -dE/dx$, where dE is the change of energy and dx is the displacement of the tip, as shown in Fig. 4a. We agree that the maximum force required with chlorine decreases slightly from ~2.0 nN to ~1.8 nN. On the other hand, the force required decreases more rapidly with the presence of the halogen atom, suggesting that the bond breaks more quickly. As to the different value of force in Fig. 3c and Fig. 3b, we thank for pointing this out. This is because different junctions were used to obtain these curves. Specifically, as shown in Fig. 3a, the molecule and the two gold pyramid tips are allowed to relax at each pulling step, which leads to Fig. 3b. In Fig. 3c, only one

electrode is used and frozen. The target molecule is also simplified to one phenyl ring (Fig. S18). Therefore, the exact values may differ due to the calculational details.

Our Revision: To avoid the misunderstanding, we have modified this sentence to “when the chlorine in TCB act on the gold atom, the force needed is slightly lower than that without chlorine (Fig. 3c)” (Page 9). We have also added the following sentence to theoretical methods in Pages 16–17: “It should be noted that the force in Fig. 3c is at least twice as large as that in Fig. 3b, because different junctions are employed. Specifically, as shown in Fig. 3a, the molecule and the two gold pyramid tips are allowed to relax at each pulling step, which leads to Fig. 3b. In Fig. 3c, only one electrode is used and frozen. Furthermore, the target molecule is also simplified to one phenyl ring.”.

Comment 4: More information on the orientation of the solvent molecule with respect to the Au atom in DFT calculations would help to understand the halogen-Au interaction. In Supplementary Fig. 18 it does not appear that there is a direct halogen-Au interaction, but the perspective might be misleading. Typical distances between Cl, I and the topmost Au atom also would be interesting to judge the strength of the interaction.

Our Reply: Thanks a lot for the reviewer’s comments. The typical distances between Cl/I and the topmost Au atom is around 2.52/2.62 Å, as shown in Fig. R9 (Fig. S19).

Our Revision: We have added the calculated model details as Fig. R9 (Fig. S19).

Fig. R9 (Fig. S19). The final states of halogen atoms (Cl and I) and the topmost gold atom.

Comment 5: In Fig. 3 of the main text, where do the discontinuities in the energy-distance dependence originate from? Is this a computational artifact?

Our Reply: Thanks for pointing out the fact of discontinuities. They are not computational artifacts. This is because the molecule and the two gold pyramid tips are allowed to relax at each pulling step. At some steps, the geometry can present significant change compared with previous step leading to some discontinuities.

Our Revision: To clarify this point, we have added the following sentence in Page 8 in the revised main text: “The discontinuities shown in the energy-displacement curve originate from significant geometry changes between two adjacent steps (Supplementary Fig. 12) after relaxation.”.

Fig. R10 (Fig. S12). The origin of the discontinuities in the energy-displacement curve. (a) The energy-displacement curve. The first discontinuity is indicated by the red dash circle. (b) The geometries of two adjacent steps corresponding to the discontinuity shown in (a). The parts with significant change are zoomed in suggesting the H and O atom in Ac group attached to gold atom resulting in a lower energy.

Reviewer 3:

General comments: I read this paper with a lot of interest. The authors studied the Au-S bond breaking reaction under halogen catalysis using single molecule junctions. A relationship among the reaction activation energy, the chemical bond breaking energy and the measured junction plateau length was established elegantly. The paper is written very well and easy to follow. The use of single molecule junctions for these type of catalysis reactions is novel and hence I feel the paper is suitable for Nature Comm. I, however, have few technical inquiries:

General Reply: We thank the reviewer very much for his/her pertinent comments, constructive suggestions and kind support. According to these comments, we have added more evidence and details, and made a comprehensive revision of the manuscript. After revisions, the manuscript has been significantly strengthened.

Comment 1: I am wondering why was not mesitylene used instead of the halogen free dodecane solvent? It appears to me that mesitylene would be a better control molecule than dodecane and more comparable to TCB and TIB.

Our Reply: Thanks a lot for the reviewer's comments. We use dodecane because it is considered more homogeneous in structure and can exclude other interferences. We have to admit that it is more rigorous to use mesitylene (TMB) as a comparison. So, we have done the STM-BJ experiment with AC-SAc in TMB solution. The results are shown in Figure R11 (Figure S5), which is consistent with the results in dodecane solvent, as only one peak of the plateau length can be observed.

Our Revision: We have done the STM-BJ experiment with AC-SAc in TMB solution. The results have been added as Figure R11 (Figure S5). The corresponding description has been added in the revised main text as follows (Page 5): "The experiment of the AC-SAc in mesitylene solution has also been done, and the results are consistent with the case in dodecane solution (Supplementary Fig. 5), illustrating that the different plateau lengths are independent on solvent viscosity."

Fig. R11 (Fig. S5). Experimental results of the AC-SAc molecule in TMB at 0.1 V. (a) 2D conductance-displacement histograms of the AC-SAc molecule. (b) 1D conductance histograms of the AC-SAc molecule. (c) Typical single conductance-displacement traces. (d) Plateau lengths of the conductance state of the AC-SAc molecule.

Comment 2: Since the authors are using Au-Au junctions and the plateau length is a central to the study, I am wondering if the authors have corrected for Au-Au snap-back distance (~ 0.5 nm) which should be added to the plateau length?

Our Reply: Thanks a lot for the good comment. Indeed, in the experiment, the Au-Au snap-back distance (~ 0.5 nm) is an objective existence. Generally, when talking about the molecular length, the Au-Au snap-back distance should be added to the plateau length. In this study, the Au-Au snap-back distance (~ 0.5 nm) has not been added to the plateau length. This is because while the plateau length is central to the study, we are concerned with the relative amount of the plateau length change during stretching in different cases, rather than the molecular length. In addition, when the Au-Au snap-back distance is added to the plateau length, it is found to be ~ 1.5 nm, which is shorter than the molecular length (~ 2.36 nm). This means that the junction breaks before the molecule is straightened to stand. Therefore, our focus is the premature rupture of the junction, which is a quantity related to the force. Considering that this may cause a confusion to readers, we have described and discussed it in the revised main text (Page 5).

Our Revision: We have added the discussion of the Au–Au snap-back distance and corresponding reference in the revised main text as follows (Page 5): “It should be noted that the plateau length in this study has not been corrected for the Au–Au snap-back distance (~ 0.5 nm)³⁶. If the molecular length needs to be determined by the plateau length, the Au–Au snap-back distance needs to be added back.”.

The following references have been added accordingly:

Ref. [36] Huang, C. C. *et al.*, Single-molecule detection of dihydroazulene photo-thermal reaction using break junction technique. *Nat. Commun.* **8**, 15436 (2017).

Comment 3: Also related to the previous question, the authors used different solvents of different viscosities. Did the author consider that different viscosities could lead to different Au-Au snap back distances, which in turn can affect the plateau lengths? A discussion about this is needed.

Our Reply: Thanks a lot for the good comment. We agree with the reviewer that viscosity may be a factor affecting the plateau length. In this study, the different plateau length peaks appear in the same solvent (in TCB case), which can illustrate that the splitting of the plateau length does not depend on the viscosity of the solvent. To further exclude the influence of viscosity, the single-molecule experiment was also carried out in TMB solvent, whose viscosity is different from dodecane. The results are added as Figure S5 in the revised Supporting Information (Page S5) and shown in Figure R11, which is consistent with the results in dodecane solvent, with only one peak plateau length of ~ 1.0 nm.

Our Revision: To exclude the influence of viscosity, we have carried out the experiment of AC–SAC in TMB solution, whose viscosity is different from dodecane. The results have been added as Figure R11 (Figure S5). The corresponding description has been added in the revised main text as follows (Page 5): “The experiment of the AC–SAC in mesitylene solution has also be done, and the results are consistent with the case in dodecane solution (Supplementary Fig. 5), illustrating that the different plateau lengths are independent on solvent viscosity.”.

Comment 4: Is there a reason why 1,2,4-tribromobenzene was not used like TIB as the catalyst to strengthen the argument?

Our Reply: Thanks a lot for the good comment. The reason why TIB is chosen as the control of TCB is because they are in two extremes, the difference is large, and the experimental phenomenon will be clearer. According to the reviewer's suggestion, we have done the experiment of AC-SAc in 1,2,4-tribromobenzene (TBB)/TCB solvent. Specifically, TBB is dissolved in TCB with a concentration of 3 mM. The results are shown as Figure R12 (Figure S16). From the experimental results, three distinct plateau lengths of ~ 0.50 nm, ~ 0.64 nm and ~ 0.98 nm can be observed, although the data quality is not as high as those obtained in TIB.

Our Revision: We have added the control experiment of AC-SAc in TBB/TCB solvent as Figure R12 (Figure S16). The corresponding discussion has been added in the revised main text as follows (Page 10): "Similarly, bromine can also be used as a catalyst to reduce the breaking energy of Au-S bonds. This results in three distinct plateau lengths of ~ 0.50 nm, ~ 0.64 nm and ~ 0.98 nm for the single-molecule junction experiment of AC-SAc in 1,2,4-tribromobenzene (TBB)/TCB solution (Supplementary Fig. 16).".

Fig. R12 (Fig. S16). Experimental results with the AC-SAc molecule in TBB/TCB at 0.1 V. (a) 2D conductance-displacement histograms of the AC-SAc molecule. (b) 1D conductance histograms of the AC-SAc molecule. (c) Typical single conductance-displacement traces. (d) Statistics of conductance plateau lengths for AC-SAc single-molecule junctions in TIB/TCB.

Comment 5: The references list is perhaps a bit brief. I recommend extending the list to include recent papers and reviews that uses single molecule junctions for chemical analysis.

Our Reply: Thanks a lot for the good comment. According to the reviewer's suggestion, we have added the references about using single molecule junctions for chemical analysis.

Our Revision: We have added the literature about the single molecule junctions using for chemical analysis in the revised main text (Page 3) as Refs. [23–27]. The corresponding description has been added as follows (Page 3): "...single-molecule junction technologies can be used for chemical analysis^{23–27}, especially for...".

The following references have been added accordingly:

Ref. [23] Dief, E. M. *et al.*, Advances in single-molecule junctions as tools for chemical and biochemical analysis. *Nat. Chem.* **15**, 600–614 (2023).

Ref. [24] Yang, C. *et al.*, Graphene–molecule–graphene single-molecule junctions to detect electronic reactions at the molecular scale. *Nat. Protoc.* **18**, 1958–1978 (2023).

Ref. [25] Zeng, B. F. *et al.*, Quantitative studies of single-molecule chemistry using conductance measurement. *Nano Today* **47**, 1001660 (2022).

Ref. [26] Nichols, R. J. *et al.*, Single-molecule electronics: Chemical and analytical perspectives. *Annu. Rev. Anal. Chem.* **8**, 389–417 (2015).

Ref. [27] Yang, C. *et al.*, Unveiling the full reaction path of the suzuki-miyaura cross-coupling in a single-molecule junction. *Nat. Nanotechnol.* **16**, 1214–1223 (2021).

Comment 6: Is there a possibility that the thioesters are reduced to thiols during the measurements? Would that affect the measurements as potentially one could have a mixture of thiols and thioesters. A brief discussion would help. All in all, this is an interesting and a clever study.

Our Reply: Thanks a lot for the reviewer's comments. We have carried out the control experiment of AC-SH in TCB solvent, whose thioesters can be reduced to thiols. Specifically, the molecule AC-SAc is deprotected by tetrabutylammonium hydroxide (TBA-OH) for 30 minutes to AC-SH. The experimental results have been shown as Figure R13 (Figure S8). Compared to the conductance peak at $\sim 10^{-4.5} G_0$ (≈ 2.451 nS) for AC-SAc, the conductance peak for AC-SH is at $\sim 10^{-4.1} G_0$ (≈ 6.156 nS), about half an order of magnitude higher, which is consistent with the reference (*Nat. Chem.* **2019**, *11*, 351) and the calculated transmission spectra (Figure R14 (Figure S9)). In addition, when the thioesters are reduced to thiols, the plateau length will be longer. Therefore, it can be determined that the splitting of plateau length peaks is not due to this.

Our Revision: We have added the experimental results of AC-SH in TCB solvent as Figure R13 (Figure S8) and the calculated transmission spectra of -SAc and -SH anchors as Figure R14 (Figure S9) in the revised Supplementary Information. The corresponding discussion has been added to the revised main text, as shown below (Page 6): "It should be noted that when the acetyl protecting groups of the AC-SAc molecules are deprotected by tetrabutylammonium hydroxide (TBA-OH) to thiols (named as AC-SH), there is only one characteristic plateau length observed (Supplementary Fig. 8). Its conductance is about half an order of magnitude higher than that of AC-SAc (Supplementary Fig. 9).".

Fig. R13 (Fig. S8). Experimental results of the AC-SH molecule in TCB at 0.1 V. (a) 2D conductance-displacement histograms of the AC-SH molecule. (b) 1D conductance histograms of the AC-SH molecule. (c) Typical single conductance-displacement traces. (d) Plateau lengths of the conductance state of the AC-SH molecule.

Fig. R14 (Fig. S9). Calculated transmission spectra of the single-molecule junction with -SAc and -SH anchors.

Finally, we really appreciate all the reviewers for their useful suggestions, patience, time and kind supports!

REVIEWERS' COMMENTS

Reviewer #1 (Remarks to the Author):

I am totally satisfied with the replies and changes made by the authors. A very neat study and impressive data quality

Reviewer #2 (Remarks to the Author):

The authors have diligently addressed the reviewers' comments, which involved conducting new experiments and enhancing the discussion and interpretation of the data. They have thoroughly addressed all aspects and comprehensively answered the referees' answers. I recommend the publication of the manuscript.

Reviewer #3 (Remarks to the Author):

The authors have made significant effort and answered all my comments. In my opinion, this work is suitable for Nature Communications and I recommend publication of the revised manuscript.

Listed below is the point-by-point response to the reviewers' comments

Reviewer 1:

General comments: I am totally satisfied with the replies and changes made by the authors. A very neat study and impressive data quality.

General Reply: We thank this reviewer very much for his/her kind support.

Reviewer 2:

General comments: The authors have diligently addressed the reviewers' comments, which involved conducting new experiments and enhancing the discussion and interpretation of the data. They have thoroughly addressed all aspects and comprehensively answered the referees' answers. I recommend the publication of the manuscript.

General Reply: We thank this reviewer very much for his/her kind support.

Reviewer 3:

General comments: The authors have made significant effort and answered all my comments. In my opinion, this work is suitable for Nature Communications and I recommend publication of the revised manuscript.

General Reply: We thank this reviewer very much for his/her kind support.

Finally, we really appreciate all the reviewers for their patience, time and kind supports!